# A 20-Year Longitudinal Study of Plasma Chitotriosidase Activity in Treated Gaucher Disease Type 1 and 3 Patients—A Qualitative and Quantitative Approach

**DOI:** 10.3390/biom13030436

**Published:** 2023-02-24

**Authors:** Paulina Szymańska-Rożek, Barbara Czartoryska, Grazina Kleinotiene, Patryk Lipiński, Anna Tylki-Szymańska, Agnieszka Ługowska

**Affiliations:** 1Faculty of Mathematics, Informatics, and Mechanics, University of Warsaw, 02-097 Warsaw, Poland; 2Department of Genetics, Institute of Psychiatry and Neurology, 02-957 Warsaw, Poland; 3Faculty of Medicine, Vilnius University, 03101 Vilnius, Lithuania; 4Department of Pediatrics, Nutrition and Metabolic Diseases, The Children’s Memorial Health Institute, 04-730 Warsaw, Poland

**Keywords:** chitotriosidase, Gaucher disease, macrophages, enzyme replacement therapy

## Abstract

Chitotriosidase is an enzyme produced and secreted in large amounts by activated macrophages, especially macrophages loaded with phagocytozed glycosphingolipid in Gaucher disease. Macrophages phagocytose decayed blood cells that contain a lot of sphingolipid-rich cell membranes. In Gaucher disease, due to a deficit in beta-glucocerebrosidase activity, the phagocytozed substrate glucocerebroside cannot undergo further catabolism. In such a situation, macrophages secrete chitotriosidase in proportion to the degree of overload. Gaucher disease (GD) is a recessively inherited disorder resulting in storage of glucosylceramide (GlcCer) in lysosomes of tissue macrophages. It is directly caused by the deficiency of beta-glucocerebrosidase (GBA) activity. Chitotriosidase has been measured systematically each year in the same group of 49 patients with type 1 and 3 GD for over 20 years. Our analysis showed that chitotriosidase is very sensitive biomarker to enzyme replacement therapy (ERT). The response to treatment introduction is of an almost immediate nature, lowering pathologically high chitotriosidase levels by a factor of 2 in a time scale of 8 months, on average. Long term enzyme replacement therapy (ERT) brings chitotriosidase activity close to reference values. Finally, reducing the dose of ERT quickly boosts chitotriosidase activity, but restoring the initial dose of treatment brings chitotriosidase level of activity back onto the decreasing time trajectory.

## 1. Introduction

Chitotriosidase (E.C. 3.2.1.14) is an enzyme produced and secreted in large amounts by activated macrophages, especially macrophages loaded with phagocytized glycosphingolipid in Gaucher disease [1]. Macrophages phagocytose decayed blood cells that contain a lot of sphingolipid-rich cell membranes. In Gaucher disease (GD), due to a deficit in beta-glucocerebrosidase (GBA, E.C. 3.2.1.45) activity, the phagocytozed substrate, glucocerebroside (GlcCer), cannot undergo further catabolism [2]. In this situation, macrophages secrete chitotrosidase in proportion to the degree of overload. However, the activation of macrophages and the secretion of chitotriosidase are also observed in other sphingolipidoses [3,4,5].

Deficiency of lysosomal GBA activity leads to Gaucher disease (GD), an autosomal recessive disorder, the most common sphingolipidosis in the Caucasian population. Three main clinical types of GD are distinguished depending on the age of onset and severity of the disease: type 1—non-neuronopathic form (MIM# 230800), type 2—severe neuronopathic form (MIM# 230900), and type 3—neuronopathic form with milder course (MIM# 231000) [6]. In rare cases, GD results from the lack of functionally active saposin C (SapC) protein, which acts as a cofactor of GBA. Patients affected with GD caused by SapC deficiency display accumulation of GlcCer and elevation of chitotriosidase activity but normal GBA activity [7].

The lack of GBA activity leads to the storage of glycosylceramide (GlcCer, called also glucocerebroside) and glucosylsphingosine (GlcSph, lyso-Gb1, lyso-GL1) in lysosomes of tissue macrophages, which become stimulated and secrete, among other compounds, chitotriosidase, which has been used for more than 20 years (since 1994) as a biomarker of Gaucher disease and an indicator of the progress of its treatment [1,8,9,10,11,12,13]. One of the limitations of chitotriosidase as a biomarker for GD is a 24-bp duplication mutation in the *CHIT1* gene which is quite common in Caucasians (about 5% homozygous and 35% heterozygous) [14]. Carriers of the 24 bp duplication exhibit nearly 50% of chitotriosidase activity compared with individuals without the duplication in the two *CHIT1* alleles.

For monitoring the enzyme replacement therapy (ERT) efficacy, other biomarkers are also used—GlcSph (the deacylated form of GlcCer) and, to a lesser extent, a pulmonary and activation-regulated chemokine (PARC or CCL18) [15]. Nowadays the experience using lyso-Gb1 in the clinical practice is still growing. It has recently been shown by our group that Lyso-Gb1 correlates with the response to the treatment measured by chitotriosidase activity only in some ranges of dosing or for splenectomized patients.

Chitotriosidase activity has been shown to correlate with several clinical parameters of GD as well as to decline in response to treatment. Chitotriosidase was the first biomarker used in GD. However, apart from a 17-year-long observation of chitotriosidase in 55 GD type 1 Romanian patients presented by Drugan et al. in 2017 [16], the literature is lacking in such a longitudinal study, with the additional feature of captured effect of dose change during the ERT treatment.

The aim of our study was to trace the changes in chitotriosidase activity in Polish GD type 1 and 3 patients on ERT during over 20 years of follow-up. We analyzed its relationship with molecular and genetic parameters drawing data from our database.

## 2. Material and Methods

The GD patients collectively consisted of 49 individuals, among whom 30 suffered from GD type 1 and 19 from GD type 3. The diagnosis of GD was based on both enzymatically proved deficient activity of GBA measured in peripheral blood leukocytes and detection of pathologic variants by DNA sequencing of the *GBA1* gene. Additionally, the presence of dup 24-bp variant in the *CHIT1* gene was searched (see below).

Patients were treated with ERT with imiglucerase (Cerezyme^®^; Genzyme Corporation, Cambridge, MA, USA), Genzyme) or velaglucerase alfa (VPRIV^®^; Shire Human Genetic Therapies, St Helier, Jersey) for a minimum of 15 years and a maximum of 28 years.

During regular (every 12 months) monitoring visits, the following examinations were performed: assessment of the clinical status, liver and spleen volume assessed by ultrasound examination, bone MRI and bone density scan by DEXA methods. A neurological examination was performed in patients with type 3 GD.

The dose of ERT was adjusted basing on disease type and clinical picture; it varied between 25 and 30 U/kg/every other week (EOW) for type 1 GD, and 50 U/kg/EOW for type 3 GD. In the observed period, there was a short restricted shortage in 2009–2010 of imiglucerase, which affected the results of chitoriosidase in all observed patients [17,18,19,20,21,22].

For each patient, in addition to the level of plasma chitotriosidase activity, we considered the following data:(a)type of GD and genotype for *GBA1* gene (detailed characteristics are provided in a Appendix A).(b)splenectomy (yes/no).(c)genotype for *CHIT1* gene (including *CHIT1* 24 bp-dup homozygote [Hm], heterozygote [Ht], and wild type [WT]).

For analyses performed, a subset of the whole cohort, satisfying a given condition, was used. For example, chitotriosidase activity was measured in only 27 patients before the start of ERT; therefore, only 27 data sets were visualized on the graph showing the short-term response to the introduction of ERT. Detailed numbers of how many patients along with the type and splenectomy status are provided for each analysis.

The study was approved by the local Bioethics Committee at The Children’s Memorial Health Institute (number 51/KB/2019), Warsaw, Poland.

### 2.1. Laboratory Analyzes

Chitotriosidase activity was measured in plasma samples by a spectrofluorometric method according to Hollak et al. [1] using the synthetic substrate 4-methylumbelliferyl beta-N–N′-N″-triacetylchitotrioside (Sigma Chemical Co, St. Louis, MO, USA). Fluorometric measurements were performed at excitation λ = 365 nm and emission λ§ = 445 nm (Hitachi spectrofluorimeter, Japan). Plasma samples were stored at −20 °C until analyzed [23].

#### CHIT1 Genotyping

Duplication of 24-bp fragment (dup24 bp) in exon 10 of the chitotriosidase gene (*CHIT1*) (rs3831317) was identified with the PCR method described by Boot et al. [14]. DNA fragments, amplified from the normal and mutant alleles, were identified after electrophoresis in 3% agarose gel. DNA samples were obtained by standard methods.

### 2.2. Statistical Analysis and Data Visualization

Gathered data is presented as chitotriosidase activity with respect to time, in different settings. Chitotriosidase activity is compared before the start of ERT, 12 to 18 months later, and 10 years after the start of ERT, with use of a *t*-test. A correction (Bonferroni type) for the *p*-values found was used to account for multiple comparisons. A simplified exponential function was fitted for every patient in order to calculate chitotriosidase activity halflife. GD1 and GD3 patients are compared in terms of response to treatment (time taken for them to achieve half of their initial chitotriosidase level). Pearson’s linear correlation coefficient is then calculated for the variables “age at start of ERT” and “chitotriosidase activity halflife”. Statistics were performed within RStudio, and pictures generated in gnuplot. Please note that for GD patients heterozygous for the *CHIT1* gene 24 bp duplication variant chitotriosidase activity level was doubled in order to normalize to the wild type (WT) *CHIT1* allele homozygous individuals, as was previously described [14,23].

## 3. Results

Time evolution of plasma chitotriosidase activity in the whole cohort of patients is presented in Figure 1 (with respect to year) and Figure 2 (with respect to the number of years on ERT), accounting for the disease GD type, genotype for *CHIT1* gene, and splenectomy. In order for the graph to be clear and pronounced, we took into account only those time trajectories that consisted of at least 10 time-points. Figure 1 and Figure 2 present the results for 46 patients, among whom:8 GD1 patients *CHIT1* 24 bp-dup Ht (2 of them splenectomised)20 GD1 patients *CHIT1* WT (2 of them splenectomised)2 GD3 patients *CHIT1* 24 bp-dup Ht16 GD3 patients *CHIT1* WT (6 of them splenectomised)

Figure 2 clearly shows that the most significant change in the activity of chitotriosidase occurred shortly after ERT introduction. The *p*-value of a paired *t*-test comparing chitotriosidase activity before the start of ERT with a measurement of chitotriosidase activity performed 12 to 18 months after the start, was less than 10^−8^. Type 3 GD patients experienced nearly a 5-fold decrease in chitotriosidase activity level within the first 18 months, whereas type 1 GD patients experienced a 3.7-fold decrease, but this difference was not statistically significant.

Next, in order to answer the question of whether the response to ERT is of an immediate or rather gradual nature, we compared the data of chitotriosidase level before the start of ERT, one year (12 to 18 months after the first administration of ERT), and ten years afterwards. Two figures visualize the decrease in chitotriosidase activity—a box-and-whisker plot with non-normalized, raw data (Figure 3), and a three-point time-trajectory with the level of chitotriosidase activity normalized to chitotriosidase level measured before the start of treatment (Figure 4). These two plots show the results for 38 patients:6 GD1 patients *CHIT1* 24 bp-dup Ht18 GD1 patients *CHIT1* WT2 GD3 patients *CHIT1* 24 bp-dup Ht12 GD3 patients *CHIT1* WT

Three paired *t*-tests were conducted to compare the means of chitotriosidase activity before the start of ERT, 12–18 months, and 10 years afterwards. The *p*-values for the tests comparing chitotriosidase level activity before the start of ERT with its level after 12–18 months and 10 years afterwards, even after a Bonferroni correction, were lower than 10^−4^. The raw *p*-value of the test comparing chitotriosidase activity after 12–18 months with 10 years after the start of treatment is 0.03, which after accounting for multiple testing, was on the of limit of statistical significance. This insight suggests that the most significant lowering of chitotriosidase activity happened within the first year of treatment, while subsequent years further lowered the activity of chitotriosidase, but in a much more steady way.

A paired *t*-test was also used to check whether GD3 patients respond better to ERT than GD1 patients (an impression one might have while looking at Figure 4 with the bare eye only). This difference is not statistically significant.

To further investigate the short-time response to ERT introduction, we gathered data of chitotriosidase activity measurements before and shortly after (up to three years) the start of ERT. For this analysis we disposed of data for 26 patients, comprising:3 GD1 patients *CHIT1* 24 bp-dup Ht (1 of them splenectomised)13 GD1 patients *CHIT1* WT (1 of them spelnectomised)10 GD3 patients *CHIT1* WT (3 of them splenectomised)

An exponential function modeling the decrease of chitotriosidase activity level in these patients has been fitted for every subject separately, i.e., we found a function $f\left(x\right)=Ae^{Bx}$ that best fits the measured activities of chitotriosidase. Time (*x*) is measured in months. Coefficient *A* is the initial value (before the start of ERT), coefficient B measures how quickly this value changes over time. Since chitotriosidase activity decreases exponentially, this coefficient is negative. The lower it is (further away from 0), this decrease happens faster.

The results of exponential fitting are presented in Table 1. For every patient, the coefficients *A* and *B* in the modeling function *f*(*x*) = *A* × *e*^*B**x*^ are given, along with the corresponding coefficient of determination *R*^2^, and the time after the start of ERT after which chitotriosidase activity reaches half of the value measured before the start of ERT. These are given by:$$f\left(x\right)=\frac{1}{2}f\left(x=0\right)$$
$$A\times e^{Bx}=\frac{1}{2}\times A$$
$$x=\frac{\ln0.5\;}{B}.$$

We call the period in which the chitotriosidase activity achieves half of its initial value (upon the start of ERT) its ‘half-life’. Please note that in the most general case the function should have a nonzero coefficient added so that it tends to a nonzero value when the argument (time) tends to infinity. However, the initial chitotriosidase activity is about two orders of magnitude greater than the value to which it tends with time (it could be, for instance, the upper norm of chitotriosidase level, 150 nmol/mL/h), so adding a constant affects the calculation of halflife to a negligible extent.

The average chitotriosidase half-life upon the start of ERT was 8 months (Figure 5). No linear correlation was found for chitotriosidase half-life and the age at start of ERT.

The conspicuous peak around years 2009–2010 visible in Figure 1 is analyzed in a close-up on years 2007–2012 in Figure 6. For these years we had 38 data vectors, i.e.:7 GD1 patients *CHIT1* 24 bp-dup Ht (1 of them splenectomised)15 GD1 patients *CHIT1* WT (2 of then splenectomised)2 GD3 patients *CHIT1* 24 bp-dup Ht14 GD3 patients *CHIT1* WT (5 of them splenectomised)

A paired *t*-test was conducted to compare the means for years 2008–2009/2010–2012. A correction (Bonferroni type) for the *p*-values found was used to account for multiple comparisons. The following pairs of variables were compared: chitotriosidase activity in 2008 with chitotriosidase activity in 2009/2010; chitotriosidase activity in 2009/2010 with chitotriosidase activity in 2012; chitotriosidase activity in 2008 with chitotriosidase activity in 2012. The respective *p*-values, even after accounting for multiple testing, were less than 10^−4^, while the third, raw *p*-value was 0.04, meaning that there was no significant difference in chitotriosidase activity in year 2008 and 2012.

This analysis allows to formulate the conclusion that lowering of ERT dose immediately affected chitotriosidase activity level, and restoring the initial dose quickly normalized chitotriosidase level likewise.

## 4. Discussion

It has been known that chitotriosidase is a sensitive biomarker of Gaucher disease that responds very well to ERT and adequately reflects the patient’s status [8,11,12,13].

Over 20 years of chitotriosidase activity measurements in the same group of patients with type 1 and 3 Gaucher disease is undoubtedly unique observation. The data allow for a reliable assessment of the value of chitotriosdase activity as a biomarker in the observation of patients with GD before treatment (ERT) and those with dosage changes; each such situation was reflected in the level of chitotriosidase activity, which proves its sensitivity as a marker monitoring the effects of ERT treatment. Moreover, our data also enabled the evaluation of chitotriosidase as a biomarker in 24 bp heterozygotes in the CHIT1 gene, a variant found in about 5% of the Caucasian population.

In the study group/cohort of 49 patients with GD, there was a group of 19 people with type 3. This allowed us to show that in type 3 GD, chitotriosidase is as self-valuable and sensitive as in type 1 GD, an observation not yet reported in the literature.

Drugan et al. (2017) provided an interesting study on 55 Caucasian patients with GD type 1 followed for almost 17 years (maximum of 7.57 and 8.96 years, before and after the initiation of ERT, respectively) (16). The most frequent genotypes (reported in 47 of 55 patients) were N370S/unknown allele, N370S/L444P and N370S/N370S. Plasma chitotriosidase activity was measured yearly before ERT and at 6-month intervals after the initiation of ERT. The mean response time, corresponding to a 50% decrease in the initial chitotriosidase activity, was nearly 6 months.

What has been shown in the analysis presented here is a quantitative proof that the response to ERT introduction (or dose reduction) is of an immediate nature. The time lag for a significant, up to 3-fold, decrease in chitotriosidase activity level might be counted in months only. Sufficiently long administration of stable doses of ERT guarantees the achievement of normal chitotriosidase activity.

The weak point of chitotriosidase as a biomarker in GD is that pathogenic variants (with commonly inherited 24 bp duplication variant) presented in one allele of the CHIT1 gene lowers its activity, wherefrom it is necessary to verify the patient’s CHIT1 genotype to correctly interpret the results of chitotriosidase activity measurement.

From a practical point of view, in the assessment of chitotriosidase activity as a biomarker of GD in the group of 24 bp carriers the value of chitotriosidase activity level should be doubled when compared with homozygous non-carrier patients.

The presented analysis is particularly valuable, from a methodological point of view, since we have been measuring chitotriosidase activity in the same group of patients with type 1 and type 3 GD for more than a decade, in some patients—even two decades. It should be emphasized that in the study period, patients had a high compliance, the dosage and infusions were in accordance with the algorithm with the exception of the shortage of imiglucerase mentioned above. What makes our cohort really unique, is the fact that all of the patients experienced a significant lowering of the dose for nonmedical reasons in 2009 [17].

Our analysis showed that lowering ERT dosage immediately affects chitotriosidase activity level [8,9] but it was quickly normalized by restoring the initial dose.

By means of statistical analysis and model fitting, we showed that the average time in which the level of chitotriosidase reached half of its initial value (from before the start of ERT) was 8 months. No correlation between the age at start of ERT and the “speed of response” to ERT has been found.

One counter-intuitive conclusion is that type 3 GD patients do not respond better to ERT introduction than type 1 GD patients, i.e., the difference in the rate of chitotriosidase activity decrease was not statistically significant between type 1 and type 3 GD.

Conversely, in the study of Drugan et al., it was found that splenectomy, bone complications and the age at treatment onset had a statistically significant impact on chitotriosidase variation. Splenectomy was performed before the availability of ERT in about one third of patients. The authors observed that splenectomized patients before ERT presented lower baseline chitotriosidase values, which declined much more slowly, as expressed by the mean response time, which was superior to 1.2 years. Likewise, in the group of patients who started ERT under 15 years of age, the persistence of elevated chitotriosidase values after the 3rd year of ERT was observed, resulting in a plateau until the end of the survey period.

Although our dataset of patients in whom chitotriosidase activity has been measured for decades is of impressive size (and diverse in terms of genotype and clinical forms), we still were not able to answer an interesting question about the role of the spleen in chitotriosidase expression. Our cohort did not have enough splenectomized patients to conduct reliable statistical tests.

## 5. Conclusions

The provided longitudinal study of chitotriosidase activity level in Polish GD type 1 and type 3 patients confirmed that chitotriosidase is a reliable biomarker quantitatively reflecting disease initial severity, its progress, and the effectiveness of treatment. The method of collecting and assessing chitotriosidase activity is relatively simple and remained the same for over 20 years, which guarantees comparability of results between years. For these reasons it is justly considered as a sensitive and useful tool for clinical practice and management of Gaucher disease patients. Its usefulness and sensitivity is comparable in both types of GD patients.

We took into account that in the case of 24 bp carriers, the value of chitotriosidase activity level should be doubled when compared with homozygous non-carrier patients.

In the present analysis we did not draw conclusions on the effect of splenectomy on chitotriosidase activity, both because the dataset of splenectomized patients was not large enough, and since in some cases the splenectomy was performed many years before the start of ERT, and in other cases, at start of treatment. Therefore, even this small group of splenectomised patients was not homogenous in term of clinical setup.

We showed that about half a year after the introduction of treatment, the value of chitotriosidase drops to half its initial (measured before the start of treatment) value and converges slowly to the limit of the upper norm (150 nmol/mg/h). This observation was independent of the type of GD and of the age at start of ERT.

The analyzed cohort of patients experienced a lowering of treatment dose, and subsequent restoring to its initial value. We showed that chitotriosidase activity level immediately responded to both these changes in ERT dosage, proving that the treatment needs to be administrated in stable doses in order to keep chitotriosidase activity on a decreasing trajectory to its upper norm.

It is not surprising to us that chitotriosidase activity assessment is still performed in many centers proving its well-established usefulness.

## Figures and Tables

**Figure 1 biomolecules-13-00436-f001:**
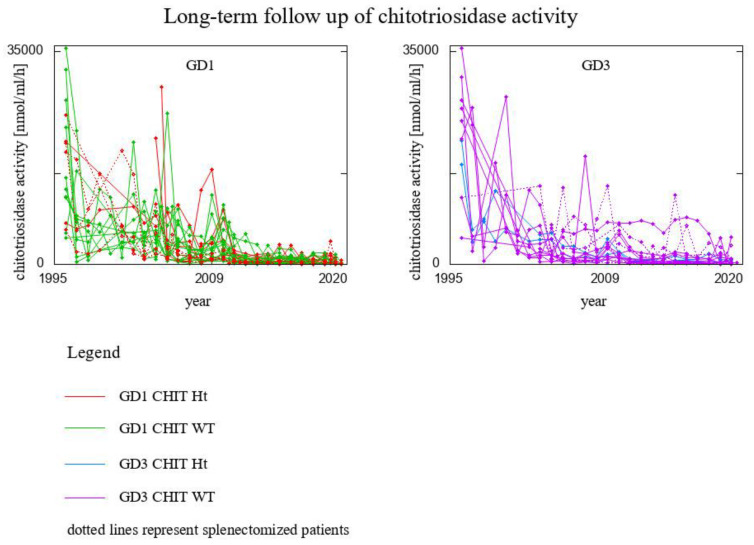
Time evolution of plasma chitotriosidase activity in the whole cohort of patients with respect to year. Data presented for 46 patients, for whom chitotriosidase level was measured in at least 10 time-points.

**Figure 2 biomolecules-13-00436-f002:**
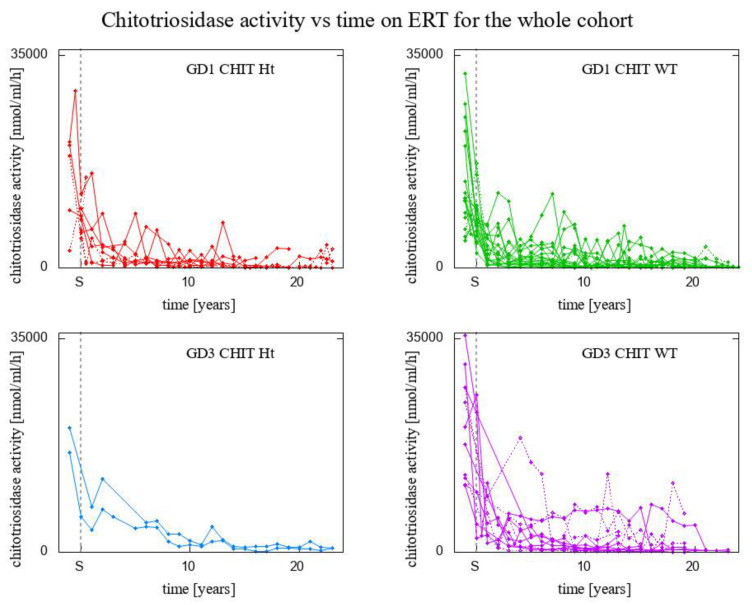
Time evolution of plasma chitotriosidase activity in the whole cohort of patients with respect to the number of years on ERT. Data presented for 46 patients, for whom chitotriosidase level was measured in at least 10 time-points.

**Figure 3 biomolecules-13-00436-f003:**
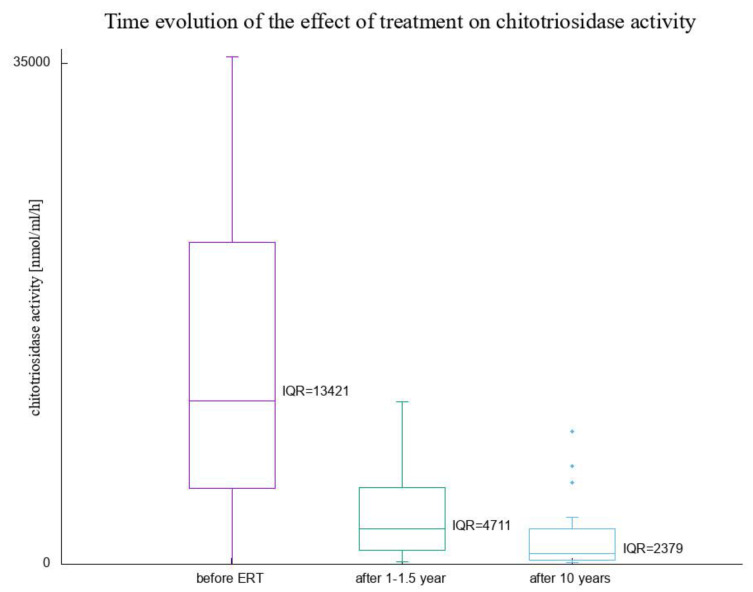
Effect of treatment on chitotriosidase activity after 1 year (12 to 18 months after the first administration of ERT) and a decade on ERT. Box-and-whisker plot presents the data for the cohort of 38 patients, for whom we obtained relevant chitotriosidase results.

**Figure 4 biomolecules-13-00436-f004:**
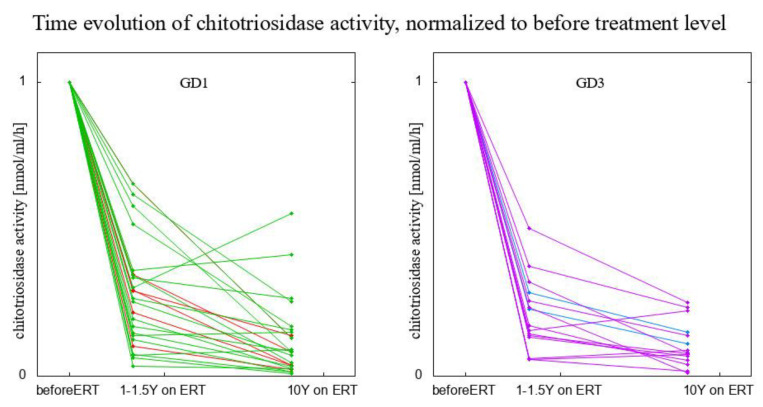
Comparison of the effect of treatment on chitotriosidase activity after 1 year (12 to 18 months after the first administration of ERT) and a decade on ERT. The three-point time-trajectory with the level of chitotriosidase activity at the start of ERT was normalized to 1, i.e., for every patient the three presented results were divided by the value of chitotriosidase activity measured before the start of ERT. Red and blue lines in each panel represent heterozygous patients for GD 1 and GD 3, respectively.

**Figure 5 biomolecules-13-00436-f005:**
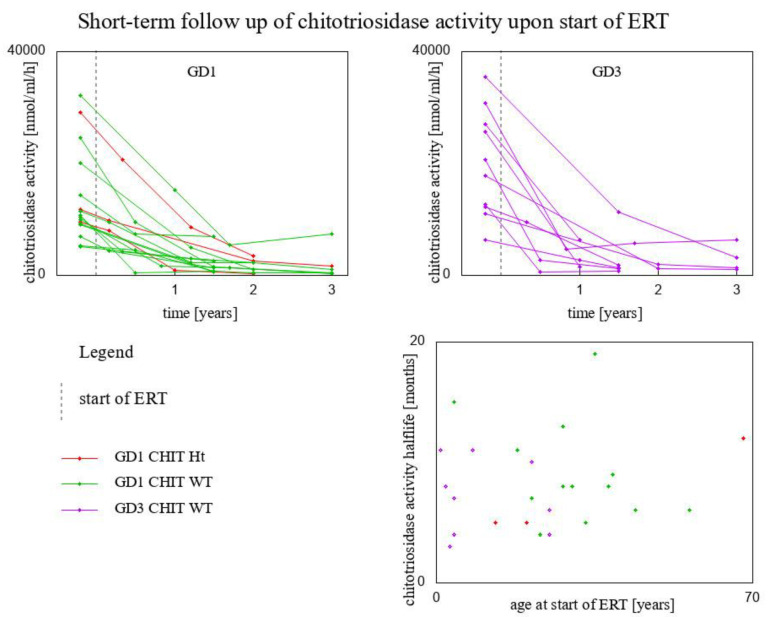
Short-time response to ERT introduction. Upper panels: chitotriosidase activity before and shortly (up to three years) after the introduction of ERT. Lower-right panel: scatter plot of the age at start of ERT and chitotriosidase half-life, i.e., the number of months it takes for chitotriosidase activity to decrease by half after the start of ERT. Results are presented for 26 patients, for whom we had the relevant data.

**Figure 6 biomolecules-13-00436-f006:**
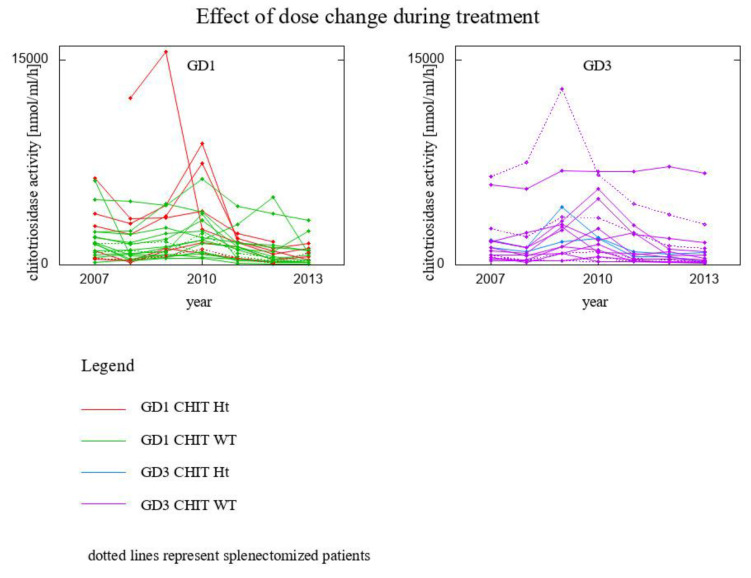
Effect of lowering the dose in 2009—a close-up on years 2007–2012.

**Table 1 biomolecules-13-00436-t001:** Results of exponential fitting of chitotriosidase activity decrease in the three years of ERT, with chitotriosidase activity half-life calculated from the fitted model.

Genotype for *GBA1* Gene/Genotype for *CHIT1* Gene	Fitted Exponential Function $f\left(x\right)=Ae^{Bx}$	$\mathrm{Coefficient}\;\mathrm{of}\;\mathrm{Determination}\;R^{2}\;\mathrm{of}\;\mathrm{the}\;\mathrm{Fitted}\;\mathrm{Exponential}\;\mathrm{Function}$	Chitotriosidase Activity Half-Life (Months)	Age at Start of ERT (Years)
Type 1 GD/*CHIT1* 24 bp-dup Ht	y = 2920 × 10^−1.061x^	$R^{2}=0.9993$	8	54
Type 1 GD/*CHIT1* 24 bp-dup Ht	y = 9560 × 10^−1.779x^	$R^{2}=0.956$	5	20
Type 1 GD/*CHIT1* 24 bp-dup Ht	y = 11,758 × 10^−0.669x^	$R^{2}=0.9876$	12	68
Type 1 GD/*CHIT1* WT	y = 20,100 × 10^−1.803x^	$R^{2}=0.9987$	5	13
Type 1 GD/*CHIT1* WT	y = 9960× 10^−1.226x^	$R^{2}=0.899$	7	21
Type 1 GD/*CHIT1* WT	y = 9280 × 10^−0.753x^	$R^{2}=0.8904$	11	18
Type 1 GD/*CHIT1* WT	y = 5292 × 10^−0.936x^	$R^{2}=0.8527$	9	39
Type 1 GD/*CHIT1* WT	y = 10,388 × 10^−1.006x^	$R^{2}=0.6497$	8	28
Type 1 GD/*CHIT1* WT	y = 24,600 × 10^−1.466x^	$R^{2}=0.9502$	6	56
Type 1 GD/*CHIT1* WT	y = 9110 × 10^−1.755x^	$R^{2}=1$	5	33
Type 1 GD/*CHIT1* WT	y = 14,300 × 10^−0.564x^	$R^{2}=0.496$	15	4
Type 1 GD/*CHIT1* WT	y = 32,100 × 10^−0.635x^	$R^{2}=0.6332$	13	28
Type 1 GD/*CHIT1* WT	y = 10,648 × 10^−2.244x^	$R^{2}=0.135$	4	23
Type 1 GD/*CHIT1* WT	y = 5180 × 10^−0.427x^	$R^{2}=0.9957$	19	35
Type 1 GD/*CHIT1* WT	y = 11,540 × 10^−1.362x^	$R^{2}=0.9992$	6	44
Type 1 GD/*CHIT1* WT	y = 6872 × 10^−1.108x^	$R^{2}=0.9556$	8	38
Type 3 GD/*CHIT1* WT	y = 6325 × 10^−0.968x^	$R^{2}=0.9801$	8	30
Type 3 GD/*CHIT1* WT	y = 35,500 × 10^−0.793x^	$R^{2}=0.999$	10	21
Type 3 GD/*CHIT1* WT	y = 12,700 × 10^−2.331x^	$R^{2}=0.3511$	4	25
Type 3 GD/*CHIT1* WT	y = 12,232 × 10^−1.257x^	$R^{2}=0.9868$	7	4
Type 3 GD/*CHIT1* WT	y = 20,633 × 10^−2.148x^	$R^{2}=0.7604$	4	4
Type 3 GD/*CHIT1* WT	y = 30,848 × 10^−0.73x^	$R^{2}=0.0328$	11	1
Type 3 GD/*CHIT1* WT	y = 11,017 × 10^−0.734x^	$R^{2}=0.9697$	11	8
Type 3 GD/*CHIT1* WT	y = 17,770 × 10^−1.059x^	$R^{2}=0.8953$	8	2
Type 3 GD/*CHIT1* WT	y = 27,090 × 10^−1.456x^	$R^{2}=1$	6	25
Type 3 GD/*CHIT1* WT	y = 25,649 × 10^−2.844x^	$R^{2}=1$	3	3

## Data Availability

All data generated or analysed during this study are included in this published article.

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
