# Peer review of "A 20-Year Longitudinal Study of Plasma Chitotriosidase Activity in Treated Gaucher Disease Type 1 and 3 Patients—A Qualitative and Quantitative Approach"

_biomolecules, 2023, doi:10.3390/biom13030436_

Round 1

Reviewer 1 Report

This is a retrospective study aiming to describe the long-term variation of circulating chitotriosidase activity in patients with type 1 and 3 Gaucher disease.
The following points need further insight or correction:
1.    All major parts of the article are numbered 1: Introduction, Material and Methods…
2.    The manuscript needs further improvement in English writing
3.    At the section Material and Methods, subchapter 2.2 Statistical analysis and data visualisation: there is no description of the statistical methods, but instead a presentation of the Results, i.e., the graphic representations and their description; this section should not include the results of the study
4.    All the data presented at the subchapter 2.2. should be included within the section Results
5.    Figure 1 is poorly understandable as it presents the evolution of the enzyme for each individual patient; I suggest it could be replaced by a representation of the regression function for each subgroup (not for each individual patient)
6.    The same applies for figures 2, 4, 5, 6 where the representation of the regression function for each subgroup would provide a better picture of the evolution
7.    At page 5, the authors indicate that figure 5 shows the enzyme changes around the years 2009-2010, but this is presented in figure 6
8.    At page 7 the authors propose a function f(x) = …, but they give no explanation for the terms used: A, B, x
9.    It is not very clear what is the usefulness of such a function that fits individual patients; the authors should propose a model of evolution that describes all patients with similar characteristics; in this respect, what is the clinical usefulness of this approach?
10.    At page 10 an explanation is provided for coefficient A, but still not for B and x
11.    The “fitted exponential function” proposed tends towards zero, but the reference values of chitotriosidase are not zero, so the function needs adjustment by adding a constant; a better approach could be to establish for each subgroup a fitted function, that can be verified against patients’ values; such functions can be useful in the clinical setting
12.    The Discussion section is too reduced, although it has interesting ideas; I believe the article can be improved if the authors will further analyse their findings and compare them to those of similar studies published on this topic.

Author Response

Review 1

This is a retrospective study aiming to describe the long-term variation of circulating chitotriosidase activity in patients with type 1 and 3 Gaucher disease.
The following points need further insight or correction:
1.    All major parts of the article are numbered 1: Introduction, Material and Methods…
Corrected as advised.
2.    The manuscript needs further improvement in English writing

The authors consulted a native speaker to improve the writing.
3.    At the section Material and Methods, subchapter 2.2 Statistical analysis and data visualisation: there is no description of the statistical methods, but instead a presentation of the Results, i.e., the graphic representations and their description; this section should not include the results of the study

As advised - information on statistical tests used, p-values considered significant, technical details on software used are now present in the manuscript, and all that can be considered as a result was moved to “Results”.
4.    All the data presented at the subchapter 2.2. should be included within the section Results.

Corrected as advised.
5.    Figure 1 is poorly understandable as it presents the evolution of the enzyme for each individual patient; I suggest it could be replaced by a representation of the regression function for each subgroup (not for each individual patient)

Figure 1 was supposed to present the data as “raw” as possible, i.e. instead of giving a table we provide a visualization of unmanipulated data, that is chitotriosidase level vs year. Data processing starts from Figure 2 on. Please note that we present chitotriosidase level evolving in time (in different settings), therefore there is no need for a regression function, which is normally used to show an interplay between two variables. Perhaps the Reviewer meant something else by “regression function”, if this is the case, please let us know.

  1. The same applies for figures 2, 4, 5, 6 where the representation of the regression function for each subgroup would provide a better picture of the evolution

Because of a huge variance within the subgroups, averaging the results in our opinion obscures the whole picture. We thought separating the subgroups in different panels and using different colours helps the eye to see the whole picture, but at the same time shows the differences between types and zygosity.

  1. At page 5, the authors indicate that figure 5 shows the enzyme changes around the years 2009-2010, but this is presented in figure 6.
    Corrected, thank you for spotting the typo.
    8.    At page 7 the authors propose a function f(x) = …, but they give no explanation for the terms used: A, B, x
    A short comment for the coefficients is now added, as well as a short discussion about the horizontal asymptote, which should not be 0, as you rightly point out in 11.
  2. It is not very clear what is the usefulness of such a function that fits individual patients; the authors should propose a model of evolution that describes all patients with similar characteristics; in this respect, what is the clinical usefulness of this approach?

The purpose of constructing the function separately for every patients was to calculate the chitotriosidase level halflife and then compare it between the subgroups. Also a correlation with age was investigated – this would not be possible after averaging.

  1. At page 10 an explanation is provided for coefficient A, but still not for B and x
    This explanation is now present when the function is introduced.
    11.    The “fitted exponential function” proposed tends towards zero, but the reference values of chitotriosidase are not zero, so the function needs adjustment by adding a constant; a better approach could be to establish for each subgroup a fitted function, that can be verified against patients’ values; such functions can be useful in the clinical setting

This is a fair point, and for the sake of scientific accurateness we redid the calculations of chtotriosidase level halflife with a coefficient C added, which we set equal to the upper norm of chitotriosidase level (150). However, because the initial values of chitotriosidase level are about two orders of magnitude greater than this upper norm, it did not change the result about chitotriosidase level *halflife*, for which this function was used. We commented on that in the manuscript.

  1. The Discussion section is too reduced, although it has interesting ideas; I believe the article can be improved if the authors will further analyse their findings and compare them to those of similar studies published on this topic.

Corrected as advised.

The provided longitudinal (over 20 years) data confirmed chitotriosidasis as a sensitive, useful for clinical practice and management of patients and also easy to perform biomarker in Gaucher disease. The relationship between chitotriosidasis activity, its dynamics of change over time and clinical features (liver and spleen enlargement), severity of the disease, as well as treatment efficacy and optimization of therapy has been shown. Chitotriosidasis activity assessment is still performd in many centres providing its usefulness.

Reviewer 2 Report

The authors are to be applauded with a rigorous analysis of plasma chitotriosidase activity levels in plasma of Gaucher disease patients prior and following therapeutic intervention. The study is concisely but sufficiently introduced. The M&M are presented in sufficient detail. The data are well presented and discussed.

The activity of the biomarker enzyme chitotriosidase in plasma samples of GD patients and normal subjects was measured in correct manner. Attention is paid to the CHIT1 genotype of investigated individuals.

The reported findings are not new by themselves (as the authors indicate as well). However, this manuscript reports in focussed manner a thorough longitudinal investigation of plasma CHIT in a real life cohort of GD patients receiving ERT. It merits publication precisely for this reason. The manuscript again convincingly reveals the importance of therapeutic GCase enzyme dose during ERT on biomarker response.

The study is conducted in straight-forward manner with appropriate methodology. No suggestions for further improvement or additions are made.

Author Response

Review 2

The authors are to be applauded with a rigorous analysis of plasma chitotriosidase activity levels in plasma of Gaucher disease patients prior and following therapeutic intervention. The study is concisely but sufficiently introduced. The M&M are presented in sufficient detail. The data are well presented and discussed.

The activity of the biomarker enzyme chitotriosidase in plasma samples of GD patients and normal subjects was measured in correct manner. Attention is paid to the CHIT1 genotype of investigated individuals.

The reported findings are not new by themselves (as the authors indicate as well). However, this manuscript reports in focussed manner a thorough longitudinal investigation of plasma CHIT in a real life cohort of GD patients receiving ERT. It merits publication precisely for this reason. The manuscript again convincingly reveals the importance of therapeutic GCase enzyme dose during ERT on biomarker response.

The study is conducted in straight-forward manner with appropriate methodology. No suggestions for further improvement or additions are made.

We thank the Reviewer for these encouraging comments and are happy to publish the results of over 20 years of observation.

Round 2

Reviewer 1 Report

The manuscript is now suitable for publication, all major problems were addressed by the authors.

Author Response

Dear Reviewer,

We are very grateful for Your suggestions which made our manuscript suitable for publication.